# Information Security and Privacy in Railway Transportation: A Systematic Review

**DOI:** 10.3390/s22207698

**Published:** 2022-10-11

**Authors:** Pablo López-Aguilar, Edgar Batista, Antoni Martínez-Ballesté, Agusti Solanas

**Affiliations:** 1Anti-Phishing Working Group—Europe, Av. Diagonal 621–628, 08028 Barcelona, Spain or; 2Smart Technologies Research Group, Department of Computer Engineering and Mathematics, Universitat Rovira i Virgili, Av. Països Catalans 26, 43007 Tarragona, Spain

**Keywords:** railway transportation, intelligent transportation systems, train, security, privacy, cybersecurity, literature review

## Abstract

Intelligent transportation systems will play a key role in the smart cities of the future. In particular, railway transportation is gaining attention as a promising solution to cope with the mobility challenges in large urban areas. Thanks to the miniaturisation of sensors and the deployment of fast data networks, the railway industry is being augmented with contextual, real-time information that opens the door to novel and personalised services. Despite the benefits of this digitalisation, the high complexity of railway transportation entails a number of challenges, particularly from security and privacy perspectives. Since railway assets are attractive targets for terrorism, coping with strong security and privacy requirements such as cryptography and privacy-preserving methods is of utmost importance. This article provides a thorough systematic literature review on information security and privacy within railway transportation systems, following the well-known methodology proposed by vom Brocke et al. We sketch out the most relevant studies and outline the main focuses, challenges and solutions described in the literature, considering technical, societal, regulatory and ethical approaches. Additionally, we discuss the remaining open issues and suggest several research lines that will gain relevance in the years to come.

## 1. Introduction

Railways are one of the most popular transport systems for both passengers and freight transportation. Therefore, strong investment in railway transportation is taking place to enhance operations and maintenance with respect to capacity, reliability, availability, efficiency, cost, safety and security. Moreover, to satisfy the growing and more demanding population, large cities and metropolitan areas are betting on public transportation (especially trains, metros and buses) to the detriment of private transport due to environmental reasons, e.g., reduce pollution levels.

With all these challenges in mind, intelligent transportation systems (ITS) [1] are gaining momentum to improve the safety, efficiency and sustainability of transport networks; minimise traffic congestion; and provide innovative and valuable services to enhance users’ experiences. Thanks to the use of cutting-edge information and communication technology (ICT), vehicles, roads, traffic signals, etc., are interconnected to improve management systems (e.g., traffic management systems and traffic signal control systems), provide efficient toll and ticketing systems, detect and alert incidents automatically, enhance guidance and assistance services and develop predictive systems based on advanced modelling techniques, among others.

In particular, intelligent railway transportation systems will be a cornerstone in the cities of tomorrow, by being fundamental in multimodal transportation and empowering emerging service models, e.g., mobility as a service (MaaS) [2]. Technologies such as the Internet of Things (IoT), ubiquitous computing, edge computing, artificial intelligence (AI) and blockchain, along with fast data networks over 4G/5G, are yet to elevate the potential of railway transportation to the next level. The progressive sensorisation of railway stations, wagons and rails is augmenting these systems, paving the way to develop the full potential of context-aware environments, such as smart stations or smart wagons. Equipped with numerous and varied digital devices, vast amounts of heterogeneous data can be collected and analysed to gather advanced knowledge and deliver real-time, effective services.

### 1.1. Security and Privacy in Railway Transportation

As the number of connected digital devices grows, the complexity of cybersecurity strategies is growing as well. Moreover, railway transportation is composed of multiple, complex and heterogeneous systems, which increasingly hinder cybersecurity management. Due to the potential number of security flaws and vulnerable protocols in such devices and communication networks, virtually every information system is prone to being attacked. Malicious actors exploit these weaknesses and are continuously seeking vulnerabilities within railway systems to disrupt their operations in the context of vandalism activities or terrorism. For instance, by targeting resource-constrained devices or insecure wireless communications, adversaries could intercept, alter, corrupt or remove railway-related data, such as operational data, signalling data and passengers’ data. Moreover, elaborated attacks and advanced persistent threats might result in attackers taking (full or partial) control of the railway system. Therefore, far from simply protecting infrastructures from occasional breakdowns, railway systems need to be prepared to face disruptive attacks, such as denial of service (DoS) or ransomware attacks.

In addition, since many autonomous decision-making processes and customer-oriented services are data-driven, they require true, accurate and complete data. In the event of attacks, the quality of the data might be compromised (e.g., integrity-related issues), so the quality of the services (QoS) provided might be degraded. Even worse, inadequate management of personal data might compromise people’s privacy (e.g., passengers data), and disclose or imply sensitive information, such as people’s physical locations, preferences or habits, due to unlawful processing.

In order to properly contextualise some information security and privacy-related issues in railway transportation, some attacker scenarios are described next, as illustrated in Figure 1. Within rail stations, attackers could disrupt numerous elementary services. For instance, vulnerable ticketing systems might allow adversaries to install malware in them to steal passengers’ banking information (e.g., credit card data) ➀ or simply disconnect the turnstiles to render impossible the access of passengers to the platform ➁. Further, insecure security checks systems (e.g., walk-through metal detector and X-ray machines) could be maliciously exploited to bypass illegal materials for terrorist actions ➂. Additionally, tampering with data or injecting fake data in the travel information display systems (e.g.,  digital timetable panels in either train stations and platforms) could cause confusion and annoyance among passengers ➃. In railway infrastructures, the use of surveillance cameras is essential, but privacy risks might emerge if such images/videos are leaked or if automatic facial recognition techniques are applied to over-surveil citizens ➄. Within rail wagons, numerous sensors and IoT devices are being deployed to capture real-time data. Given their generally resource-constrained nature, attackers could exploit their vulnerabilities to inject fake data or interrupt the sensor networks ➅. Additionally, long-distance train wagons are increasingly integrating novel, personalised entertainment systems to make travel more pleasant. If these systems are unprotected, malicious actors could steal sensitive data about passengers, such as their preferences ➆. Similarly, privacy risks could also arise when passengers post travel information to social networks ➇. Concerning the communication among the different systems, security issues are likely to appear if they use vulnerable protocols or obsolete technologies. Attackers could launch multiple attacks, such as DoS, jamming or man-in-the-middle attacks, to gain access to the system or disrupt the proper functioning ➈. Last but not least, decision support systems, which rely on legitimate data and numerous autonomous processes, are susceptible to poisoning attacks (in case of using AI techniques) with unpredictable consequences ➉.

To this end, the need for cybersecurity frameworks, standards and policies is undeniable, and the adoption of up-to-date security protection countermeasures is first and foremost. Notwithstanding, raising awareness is essential: it is crucial to ensure that railway organisations, manufacturers, decision makers and stakeholders are aware of novel cyberthreats. To this end, cybersecurity training programs and the assessment of digital skills are cornerstones of security at a global scale.

### 1.2. Contribution and Plan of the Article

Despite the large number of security/privacy-related studies [3,4,5,6,7], their applicability to railways is often disregarded. Instead of considering these aspects afterthoughts, railway systems must be developed following a security-by-design approach. Given the large complexity of railway systems, reviewing all the individual concerns in each of them is a daunting task. In consequence, to help readers properly understand all the challenges involved, this article adopts a more neutral and high-level approach, by focusing on them not only from a technical perspective, but also from societal, regulatory and ethical perspectives. Specifically, the aim of this article is to provide a timely review of the state of the art in the field of information security and privacy for railway transportation. The contribution of the article is threefold: (i) reviewing the current research done in this field following a strict review methodology; (ii) classifying and summarising the research in terms of focus, challenges and opportunities; and (iii) identifying the main research gaps to pinpoint future research directions. All in all, we hope that this article helps technicians, practitioners, stakeholders and researchers to set the grounds for more secure and private railway transportation systems.

The rest of the article is organised as follows: Section 2 presents the review methodology; Section 3 summarises the results of the review in accordance with our taxonomy; Section 4 provides an extensive discussion on the development of security and privacy practices; Section 5 describes the limitations of our work; and Section 6 summarises our conclusions and points out further research lines that are expected to be relevant in the years ahead.

## 2. Research Methodology

Our review methodology was based on the five phases of vom Brocke et al. [8] for a systematic literature review. More precisely, the phases are the following: (i) definition of the review scope, (ii) conceptualisation of the topic, (iii) literature search, (iv) literature analysis and synthesis and (v) definition of a research agenda.

### 2.1. Definition of the Review Scope

We aimed to extract and analyse the current state of the art in techniques, protocols and technologies related to information security and privacy for railway transportation. More specifically, the scope of the review is defined according to the taxonomy proposed by Cooper [9]:Focus: To gain a broad view of the field, the literature review focuses on all types of academic articles, ranging from theoretical approaches to more practical solutions.Goal: This review investigates which procedures and solutions have been proposed by the scientific community to address the security and privacy aspects of railway transportation, which are the challenges identified in the field, and which are the future research directions.Organisation: The review is organised using a conceptual structure, i.e., grouping similar ideas from different authors in order to easily guide the reader through the topic.Perspective: In this review, we adopt a neutral but critical position regarding the articles studied from the literature.Audience: It is intended for researchers, practitioners and stakeholders in the field of railway transportation, information security and/or privacy protection.Coverage: This literature review provides exhaustive coverage of the available, published scientific literature.

For the sake of completeness, our approach also relies on several research questions pertinent to cyber-security and privacy protection, which are aligned to the specific objectives of our article (see Table 1). Based on these research questions, we performed a thorough analysis of the available literature and analysed the most well-known solutions, along with their challenges.

### 2.2. Conceptualisation of the Topic

According to vom Brocke et al. [8], each review must clearly provide a broad conception of what is known about the topic and potential areas where knowledge is needed. This review addresses two key topics: (i) railway transportation and (ii) information security. On the one hand, railway transportation refers to the transport of passengers and goods on vehicles running on rails, such as trains, metros and trams. More specifically, in this review, we concentrate on those vehicles, infrastructures and IT systems (e.g., ticketing systems and surveillance systems) involved in passenger transportation only. On the other hand, information security often refers to the preservation of the confidentiality, integrity and availability of information (i.e., the CIA triad). Additional properties, such as non-repudiation, authenticity, trustworthiness, accountability, auditability and privacy, are considered as well [10,11,12]. Therefore, for the sake of completeness, we consider all these properties to broaden the scope of information security. Even though privacy is already embedded in the concept of information security, the fact that it is recognised as a human right makes it a sufficiently important topic to justify a distinctive analysis. For a complete contextualisation of the synergies between both topics, we refer the reader to Section 1.

### 2.3. Literature Search

Different steps are involved in this phase—namely, database selection, keyword searching and backward and forward searching—alongside the ongoing evaluation of the sources. These steps, described below, are also summarised in Figure 2.

#### 2.3.1. Database Selection

To properly guarantee the coverage of high-quality scientific literature in our review, the well-known Web of Science database was used.

#### 2.3.2. Keyword Search

Since the scope of this review is multi-faceted, numerous keywords have been chosen to build the search string. On the one hand, a number of railway transportation related keywords were selected to carefully describe all sorts of vehicles and transportation means. On the other hand, all the dimensions related to information security identified in the previous phase were also included. With the aim to obtain the widest possible coverage on the topic, we did not apply any timespan criteria to our search, and the search string was configured to seek within all the article (i.e., words in the title, abstract, keywords, full-text and metadata). The resulting search string was the following:


ALL ( (“transport” OR “transportation”) AND (“railway” OR “train” OR “wagon” OR “rail” OR “railroad” OR “subway” OR “metro” OR “tube” OR “underground”) AND (“security” OR “privacy” OR “confidentiality” OR “integrity” OR “authenticity” OR “availability” OR “non-repudiation” OR “accountability” OR “auditability” OR “trustworthiness”) )


#### 2.3.3. Literature Evaluation

The eligibility of the retrieved literature was evaluated based on a set of inclusion and exclusion criteria for quality assessment. In particular, the following inclusion/exclusion criteria were applied:$IC_{1}$. The publication was written in English.$IC_{2}$. The publication was peer-reviewed (to avoid grey literature).$IC_{3}$. The full-text of the publication was available.$IC_{4}$. The publication was published in a Q1 journal (according to the ISI-JCR) or its number of citations was above the 75th percentile regarding the citations of other articles published that year.$IC_{5}$. The publication was relevant to the subject. In this review, relevancy means that the publication contextualises information security and privacy aspects in railway passenger transportation, and that the keyword terms are properly used in their lexical context. This excludes publications in which the terms only appear in the references section, are tangentially mentioned, or result from typos.

The selection process was performed in two stages. First, publications that did not meet criteria $IC_{1}$ to $IC_{4}$ were excluded from the review from an objective perspective. Second, we conducted multiple screening phases to fulfil criterion $IC_{5}$, according to the article’s title, abstract and full-text. During the screenings, each article was labelled as accepted or rejected. In order to lessen researcher bias, a cross-checked evaluation was conducted among the four researchers, each of them independently classifying the articles according to the aforementioned criteria. For the sake of completeness, the screening followed a conservative approach: a publication was preliminary accepted if it had, at least, one favorable assessment. Conversely, a publication was rejected when it did not get any positive vote. This work was carried out using a shared spreadsheet, which contained all relevant information of each publication (e.g., title, authors, year, …) and the reviewers assessments (e.g., vote and reason).

#### 2.3.4. Backward and Forward Search

From the initial selection of articles, we discovered additional studies using the so-called backward search, i.e., reviewing older literature cited in the selected articles, and forward search, i.e., reviewing articles that have cited the selected articles. Each search produced a new set of publications, which we evaluated again following the procedure described in Section 2.3.3. This iterative approach enables a more in-depth literature search, thus endowing this review with a major robustness.

### 2.4. Literature Analysis and Synthesis

To address discrepancies among researchers in the previous phases, several round-table meetings were conducted to discuss whether those articles with non-unanimous positive votes fulfilled the inclusion criteria. After, a poll was conducted and any article not achieving a majority of votes (>50%) was excluded from the review.

Accepted articles were carefully analysed and all relevant information was extracted, characterised and classified. With the aim to generate new knowledge grounded on the selected articles, a conceptual synthesis approach was chosen so as to identify common topics across the articles. Therefore, this review presents the results from multiple perspectives, e.g., according to the scope or the technologies used.

### 2.5. Definition of a Research Agenda

The purpose of this review is not only to survey what has been studied in the field, but also to provide solid foundations for further research on the topic. Based on the results of our study, further research lines are outlined in Section 6.

## 3. Results

First search was queried on 9th February 2022, and all publications indexed to the date were retrieved. All the aforementioned keywords were searched in all article’s fields (e.g., title, abstract, keywords, full-text, metadata, …). As a result, a total of 3913 references were obtained. No duplicated results were retrieved because a single database was queried. From those, 1219 references remained after applying the objective inclusion criteria ($IC_{1}$ to $IC_{4}$). Given the large set of remaining references, a preliminary title screening was conducted to discard articles that were clearly out of the scope (e.g., not related to transportation), and 966 references were excluded at first. Afterwards, 211 references were excluded based on the abstract screening. Hence, a total of 42 references were assessed for eligibility. After full-text screening, 14 references were rejected and 28 publications were accepted in first search.

From those 28 accepted publications in the first search, a forward search was conducted on 18th March 2022, where 398 references were returned. We excluded 211 references that did not meet the inclusion criteria $IC_{1}$ to $IC_{4}$, plus 10 references that had already been accepted in the first search. Next, 177 references were screened following the same procedure. As a result, 168 references were excluded after abstract screening, and 9 references were assessed for eligibility. From those, 3 references were excluded after full-text reading, and 6 publications were therefore, accepted to be included in our review.

Finally, a backward search was conducted on 31st March 2022, resulting in 645 references. Then, we excluded 308 references that did not fulfil inclusion criteria $IC_{1}$ to $IC_{4}$ along with 2 additional references that were already accepted in the first search, thereby remaining in 335 distinct references to be screened. A total of 324 references were excluded after abstract screening, and another 6 references were excluded after full-text reading. Therefore, 5 publications were accepted at this stage.

With the aim to enhance transparency, we provide the scientific community with the individual assessments made throughout this process. Spreadsheets are available from our research group website (https://smarttechresearch.com/opendata/sensors2020/Screening.zip. Accessed on 29 August 2022. SHA-256 hash: c76e8e802439ba8d22f2be2661956e1c81c88b 86c3485381fab3ff09145d433d).

The selected 39 articles included in this literature review are listed in Table 2. More specifically, 33 of these articles are published in journals, 5 are published in magazines and only one is published in a conference. The leading journals in this review are IEEE Transactions on Intelligent Transportation Systems with 5 articles, followed by IEEE Internet of Things Journal with 4 articles, and both IEEE Communications Magazine and IEEE Access with 3 articles each. Consequently, IEEE is the most relevant publisher within this scope with 24 articles, followed by MDPI with 4 articles and Elsevier and Springer with 3 articles each. Finally, Wiley (2 articles) and Emerald, PLOS and SAGE (1 article each) have less relevance (see Figure 3a). From a temporal view, a growing tendency of relevant articles in the field is noticeable since mid 2010s (see Figure 3b). Indeed, more than the half of the articles analysed in this review have been published in the last three years. From a geographical perspective, this research is more actively investigated in China with 47 authors, followed by UK with 20 authors, Spain with 15 authors and Sweden with 11 authors. The greatest concentration of articles is from Asia, especially in China, Europe, North America and Australia (see Figure 3c). Finally, to preliminary scan the articles, Figure 3d illustrates the most relevant keywords (i.e., index terms) of the selected articles. As expected, the most relevant keywords are, among others, railway, train, transportation, cybersecurity, security and privacy, which are aligned with the search string keywords and the scope of this review. In addition to that, this practice helps identify further keywords that will also be relevant in the selected articles, such as blockchain, WSN, machine learning, ETCS and CBTC.

Articles have been classified into three groups based on their scope and goals, as follows:Enhanced systems for increasing safety and security in railways: Articles proposing techniques, methods or tools to improve the railway infrastructure from an information security or privacy perspective.Cybersecurity issues and challenges in railways: Articles reviewing or describing issues and/or challenges related to information security or privacy in railways.Users’ cybersecurity awareness in railway infrastructures: Articles addressing or evaluating the awareness of users on information security or privacy in the railway infrastructure.

Nearly half of the articles are encompassed within the first group (20/39), whilst the second group and the third group comprise 15 articles and 4 articles, respectively, as observed in Table 3. Furthermore, as observed in Figure 4, we can notice a tendency shift regarding the types of articles published in this field throughout the years. Whereas initial studies mainly discussed issues and challenges in the field, most of the recent articles are more technical by proposing solutions to increase the security of railways. Therefore, an evolution from theoretical to practical approaches is highlighted. Likewise, cybersecurity awareness-related studies are likely to continue increasing in the years to come. Each of these three groups is explored individually in the subsequent sections.

### 3.1. Enhanced Systems for Increasing Safety and Security in Railways

Over the years, railways have been massively used by billions of passengers. Moreover, its use for short or long displacements is nowadays far from decreasing. Consequently, governments and manufacturers are investing in developing new applications aimed at guaranteeing travelers safety based in physical and digital perspectives (e.g., protect passengers data when traveling by train, improve decisions based on trains digital data, etc.). Hence, the development of systems to guarantee passengers data protection from technical, ethical and legal perspectives remains a key challenge in the near future. Thus, besides providing thorough information on the state-of-the-art of data control systems and describe some of the most relevant cyberthreats, many literature suggests improvements to build safer, more secure and efficient data management systems in the railway industry.

*Secure and reliable self-monitoring.* All trains should constantly and reliably perform self-monitoring actions, and report their position without any infrastructure support. Based on the above, with the aim to make railway more efficient and secure, the work in [31] provides a WSN-based topology, communication protocol, application and sensor nodes prototypes designed for low-power timely train integrity reporting in unreliable conditions. Train Collision Avoidance Systems (TCAS) was enhanced with four sensors that redundantly monitor train coupling, and send the data to a control centre that receives the reports. However, the proposed prototype should improve the reliability and energy efficiency, and perform tests on real environments. The study employs experimental tests to perform qualitative checks of the network operation and stability in different operation conditions. Additionally, focused on the improvement of wireless networks, Sun et al. [42] proposes a novel train control scheme to enhance the QoS of communication-based train control systems (CBTC). The scheme implements a retry limit adaptation at the medium-access control sublayer to minimise the energy consumption. Additionally, it provides guidance trajectory update to mitigate the trip time tracking errors. Apparently, in comparison to existing schemes, the proposed solution could improve the energy efficiency in CBTC systems. According to the study, further research should consider variables like train speed, and other wireless networks technologies like WiMAX and LTE-Advanced. Relatedly, focused in CBTC and aiming to boost cyberdefences to avoid potential train collisions, Kim et al. [28] proposes a countermeasure structure for improving the security on train control systems. In particular, the authors implement a realistic CBTC testbed on a commercial train control and supervision software to capture the most relevant CBTC components. Apparently, the positive analysis of the results confirms the validity of the system.

*Efficient resource management.* Being ITS a major factor in the smart cities of tomorrow, ICT offer numerous opportunities to provide interactive and time-efficient services and to improve urban mobility. To this end, Duan et al. [20] proposes a framework to optimise the operation and management of smart cities based on an effective ITS model for metro and electric vehicles. In particular, the study aims at enhancing the security of data transaction within the smart city, and provides an improved secure data transaction framework based on directed acyclic graphs. Given the complexities of the numerous inter-dependencies within the railway network, the work in [34] provides a review of the requirements and challenges in applications using big data in railway asset management systems. According to the study, further efforts aiming to provide common standards and frameworks are required to facilitate data management and improve decision-making processes.

*AI-based applications.* With a practical implementation of deep learning methods for increasing safety and security in railway crossings, the authors in [41] propose an AI-based surveillance system for railway crossing traffic, called AISS4RCT. This system is based on a combination of detection and classification methods such as You Only Look Once (YOLO) [52] that achieves an average recall of 89%, the authors state. The study focuses on various image processing inputs, including vehicle presence, pedestrian presence, vehicle trajectory tracking, railway barriers at railway crossings, railway warnings and light signaling systems. Moreover, the detection of dangerous situations at railway crossing in real-time was performed using accelerated image processing techniques and deep neural networks. Additionally, leveraging AI techniques in cyber-physical systems (CPS), i.e., interconnected software components operating on different scales and devices, the work in [25] presents an administration framework to protect cybersecurity, privacy, and dependability in railways. The framework aims at monitoring and managing real-time CPS by integrating an innovative secure routing protocol with a policy-based authorisation features covering confidentiality, integrity, and authentication. According to the study, the system is energy efficient and could effectively boost defence systems against a large volume of threats. Additionally, the simulations report an average delay in mitigating attacks (under normal operation ranges) from 0.2 to 0.6 s. A secure integration with external systems is considered in future work. Furthermore, based on a decentralised collaborative ML approach, [40] introduces a privacy-preserving location recommendation framework that keeps users’ data on their devices. Although the framework has some limitations (e.g., long time to train the model, high need of computing resources…), the offered method could be considered as an important step towards the implementation of privacy-preserving AI technologies. Additionally, using ML to improve decision-making processes for the security and safety of railway stations, Alawad et al. [14] offers an overview of WSNs along with different applications that could be used in smart railway stations. Interestingly, the study highlights the need to monitor individual behaviour to implement more secure WSNs. According to the authors view, further research in AI will enable more efficient data management in railway stations.

*Blockchain-based applications.* Focusing on the enhancement of railway management services, Mu et al. [36] provides a blockchain-based scenario using a set of policies where users signing keys were associated with a policy set that restricts users’ rights. In terms of the signature generation cost, the proposed scheme has, seemingly, lower computational time compared to other schemes [53,54]. Although the study only provides experimental results, the applicability of a secure policy driven signature scheme with blockchain might be an interesting alternative to existing systems. Moreover, the study in [23] analyses the application of blockchain-based technologies for the implementation of e-Tickets, traceability, asset management, security, and privacy. In particular, authors propose an alarm collection system based in Ethereum that ensures traceability and privacy of the alarms. Additionally, aiming to guarantee the participation of any railway in the network, the system implements an Ethereum blockchain to ensure information security and data privacy. Similarly, [49] implements a framework using blockchain technology to encrypt data transmitted among different energy systems in smart cities. In this framework, each energy exchange was considered as a block encrypted using a hash function to provide encryption and protection against cyberattacks. This platform aims at improving the interoperability of transportation technologies, namely, vehicle-to-grid (V2G) and vehicle-to-subway (V2S).

*Secured cloud applications.* According to [51], “Under the cloud environment, information security is critical to ensure the integrity of data through protection from unauthorised manipulation and the confidentiality through protection against the leakage of sensitive information”. Based on the previous assumption, the study proposes a lightweight authenticated encryption scheme, and uses associated data to provide confidentiality and integrity in the railway IoT-cloud environment. Aiming to homogenise the management of large volumes of data from different sources, Dong et al. [19] experimented with cloud architectures. More specifically, the study proposed a new architecture for a secure vehicular cloud computing system for intelligent high-speed railways (HSR), called SVCC-HSR. Thus, to ensure the security and efficiency of communications, the architecture implements three key security mechanisms, namely, fast authentication, hierarchical data encryption and efficient utilisation of unpredictable resources. According to the results, the system improves the management of large volumes of data with more efficient utilisation of unpredictable resources.

*Robust balise systems.* As previously stated, the large volume of interconnections inside railway systems, and security and resiliency in CPSs, must be correctly tackled. The works in [21,45] aim at improving defence strategies against cyber-physical attacks in balise systems. Balises are electronic components placed between the rails that allow vehicle–ground communications based on radio frequency. These devices, which constitute an integral part of an automatic train protection, allow a plethora of applications to improve the security of infrastructures. For instance, the ETCS-compliant balise, called Eurobalise, sends messages to the trains (through their balise transmission module, hereafter BTM) comprising its location, rail gradient and train speed limit. The work in [21] implemented ML methods capable of detecting attacks and failure operations in these systems. Although the proposed method requires training data to detect threats, the simulation showed noticeable accuracy in railway systems. Additionally, a fuzzy system was developed as a countermeasure to improve train safety against cyberattacks. Relatedly, the lack of cryptographic protection in balise communications led [45] to performing several attacks and providing their countermeasures. Whilst the study was not performed in real environments, the simulation was based on a real subway line configuration.

*Enhanced and secured communications.* Video entertainment programs in railway stations can bring better experiences to passengers, but the wireless technology used by these systems is a target for cyberattacks. With the intention of preventing them, Wu et al. [47] presented a scheme to ensure the code stream’s authenticity and enable the vehicle to improve the passengers’ experiences by providing high QoS in the subway WiFi environment. The proposed scheme incorporates an operation centre that creates a trusted scalable video code (SVC) code stream broadcast to all the access points (APs) on the subway lines. Thus, based on variables such as location, speed and wireless signal quality, each AP adjusts the WiFi bit rate and truncates the code stream to enable passing trains receive the high-quality video in real time. From a different perspective, but also aiming to increase the security and safety in the railway sector, the study in [13] proposes a mobile network architecture that enables high-data-rate wireless connectivity. Moreover, the architecture could be implemented in different scenarios, namely, inter-car, intra-car, inside station, train-to-infrastructure and infrastructure-to-infrastructure. To achieve the requirements, the study proposes a massive MIMO-based wireless coverage for train stations and rail cars, and discusses the technical challenges associated with the implementation of these techniques.

Based on improving security and efficiency in Long Term Evolution for Railway (LTE-R), Wang et al. [44] analysed the vulnerabilities of the LTE-R access authentication protocol, and proposed a novel scheme based on proxy signature authentication to enhance its security while preserving its efficiency. The study proposes three main security mechanisms: (i) a novel elliptic curve cryptosystem-based certificateless proxy signature designed for authentication security, (ii) a hash-based puzzle for protecting the system against DoS attacks and (iii) a key pre-generation mechanism used to improve the efficiency of fast-handover authentication. Additionally, to enhance protection against these attacks in the communication network of high-speed trains, the authors of [50] provided a secure control framework that enables restoring the network after DoS attacks. The proposed control scheme contains an attack detection strategy, a communication recovery scheme and distributed controllers to provide resiliency to the system.

### 3.2. Cybersecurity Issues and Challenges in Railways

The advent of data-driven decision-making algorithms in the railway industry is bringing new ways to efficiently manage operations, provide reliability and maintenance and explore future improvements in passenger experience. However, the large complexity of railway transportation systems entails several issues and challenges. Many authors are exploring these aspects, and proposing novel insights and research directions for further work.

With digitalisation, cybercriminals found unprecedented opportunities to perpetrate their malicious actions. In this context, Thaduri et al. [43] provided a comprehensive overview of the most relevant threats, challenges, vulnerabilities and risks in the railway infrastructure. Moreover, the study provides different cybersecurity frameworks that can help enhance data protection systems. Similarly, Kour et al. [29] provided a timeline review of the most relevant cybersecurity incidents in railways, and highlighted the main challenges that should be addressed, such as the increasing use of cloud and IoT technologies, and the ability to counter cybersecurity attacks. Comparably, Zeng et al. [48] provided valuable information on the challenges and technologies associated with transportation infrastructures. Although the article was published more than a decade ago, most of the technologies mentioned (e.g., video surveillance, tracking and location, authentication and access control) are still relevant in the field today.

A number of articles have explored communication networks in the context of railways. The study in [24] delves into the requirements for deploying such networks, and introduces the most relevant technologies in the area. Likewise, it proposes several fields where enabling technologies, such as AI, big data and sensors, can potentially revolutionise the railway industry. In order to foster the European Rail Traffic Management System (ERTMS), the work in [32] provides a security analysis and presents some recommendations. Among other countermeasures, a more robust cryptographic mechanism, the implementation of a new key distribution scheme and a new key storage and new system integrity module, would harden the defences against some of the most current relevant threats. Likewise, Ma et al. [33] provides countermeasures to protect wireless communication systems from jamming attacks. According to the authors, the deployment of a smart monitoring system and position-aware assisted smart antennas contributes to preventing these attacks. Focused on wireless technologies also, the study in [26] provides a throughout survey on WSNs applicable to the railway industry. The article brings a practical approach on the different types of sensors and networks to develop multiple use-case scenarios. Similarly, the work in [22] presents a survey on the evolution of communication technologies for railways. Furthermore, the article discusses the reasons behind the use of radio and WiFi in CBTC systems, while presenting the main requirements and the most common standards. Finally, Moreno et al. [35] explains the opportunities and challenges that should be addressed in radio technologies. On the one hand, the article focuses on services intended to increase safety in railways (e.g., signalling services), and on the other hand, the authors discuss the challenges and opportunities related to operational non-safety services (e.g., Internet access).

Additionally, cyberattacks targeting industrial systems are becoming more sophisticated and prevalent. For example, intrusion into security information and event management systems (SIEM) can lead to incidents with serious consequences. Relatedly, the work in [17] discusses an approach to detecting abnormal activity within these systems to develop more effective incident detection techniques.

Being one of the most crowded places in large cities, train stations also pose tremendous challenges from a security and privacy perspective. Tracking pedestrian movements in real-time in such uncontrolled public spaces implies tremendous challenges in terms of computational cost, object recognition and privacy issues. With the aim of improving efficiency, the work in [39] proposes a framework for representing pedestrian-to-pedestrian interactions using vector-weighted graphs to analyse the physical distances among individuals. This method, tested on a train platform, was intended to identify offenders in train stations in a real-time, privacy-friendly fashion. Concerning human behaviour also, the studies in [18,38] analyse whether security, privacy and liberty aspects affect the desire to travel by rail: it is shown that passengers might prioritise security improvements to the detriment of their privacy and liberty, e.g., by adding surveillance cameras. Aiming to preserve privacy too, the work in [27] provides methods to find the most effective surfaces for speech isolation in high-speed train wagons.

Finally, concerning balise systems, Wu et al. [46] simulated the vulnerabilities between a balise and a train’s BTM, such as the possibility of malicious tampering of the air-gap communication channel, and proposed countermeasures to enhance the security of the ETCS.

### 3.3. Users’ Cybersecurity Awareness in Railway Infrastructures

The implementation of training and cybersecurity awareness campaigns remains an important step towards maximising potential victims’ resilience. Consequently, the study in [30] provides an estimation of the cybersecurity maturity and awareness risk for workforce management in railways. Moreover, the paper provides some recommendations and literature aiming to enhance the cybersecurity workforce culture in the industry. Comparably, Bellini et al. [15] presents a formal definition aiming to translate cyber resilience into an operational tool. In particular, starting from a holistic approach, the paper proposes a methodology to unify the different security matters existing in the railway domain, and introduces a novel cyber-resilience domain model.

Smartphones provide passengers with a variety of mobile applications with which to manage and purchase their trips more easily. However, most of the passengers are not aware that many of these applications generate data that can be used to enable malicious actions. In the context of massive use of mobile phones, and based on a recent experiment performed at the Melbourne train station, the work in [16] provides information on the attitude of passengers towards location data collection activities. Further, given the increase in sensor-based monitoring systems, the study highlights the need to undertake awareness campaigns to inform users about their location privacy and its potential consequences. Additionally, in the context of privacy awareness, Patil et al. [37] highlights the challenges associated with the increasing volume of data led by the implementation of new applications in the transportation industry. More precisely, the study presents a pan-European experiment to assess respondents’ perceptions with regard to security and surveillance in European railways.

## 4. Discussion

The articles summarised in the previous section provide information of the cyberthreats and bring different solutions to address some of the most relevant challenges in the railways sector. Relatedly, and under the scope of the NIS Directive [55], the report [56] published by the European Union Agency for Cybersecurity (ENISA) presents the relevant trends and identifies several challenges to overcome in the near future. Thus, based on the articles assessed throughout this study and in accordance with the report, this section discusses some of the most significant technical and societal aspects that might be addressed to improve the current cybersecurity gaps in this industry. Likewise, the relevant technologies and architectures that could potentially enhance the railway cyberdefences are also stressed. Finally, this section highlights the need to provide solutions to the current lack of interoperability in data management systems, and provide more specific regulations addressing legal and ethical issues.

### 4.1. Technical Aspects

The large number of different environments, the high consumption of energy, the traffic flow and the endless safety and security risks existing in the industry, foster the need to adopt new systems employing technologies such as WSNs. Thus, the implementation of this technology might not only bring energy-optimised configurations [31], but also enhance security, safety and decision-making processes, and facilitate systems integration [14]. Additionally, the use of balises to continuously report train positions to ensure their integrity can be seen as a potential attack vector for criminals due to the devastating consequences of a cyber-physical attack. Although several studies have mentioned these vulnerabilities, the implementation of a challenge–response authentication process [46], the use of software programs based on AI to detect anomalies [21] or the use of cryptography to protect the communication between different parties [45] can bring interesting solutions to existing threats. Notwithstanding, the implementation of new technologies in railways should always be performed following relevant guidelines for cyber risk management [57]. Figure 5 lists the number of articles aiming to provide solutions to the most common attacks targeting railways.

Over the years, the use of GSM-R has been widely adopted in the railway industry. However, its poor transmission rate per connection and its packet delay might not be sufficient to cover the current demand brought by the digitisation of the sector [58]. Thus, further research should seek to provide new communication technologies (e.g., LTE-R) and cover the needs brought about by the advent of new train services. The implementation of these technologies will, undoubtedly, bring several advantages and enable the use of new applications, and therefore improve passengers’ safety and their perceived QoS. Nonetheless, this will also imply higher costs and increase the already existing problem of interoperability. Relatedly, despite the efforts devoted by the ERTMS to ensure interoperability of the railway systems, the implementation of new devices connected to the network might lead to increasing difficulties to homogenise data management and improve the security in railways. Hence, future efforts should provide common standards addressing interoperability issues while improving security and efficiency in decision-making processes. Relatedly, Figure 6 lists the articles described in Section 3.1, along with the security and privacy-by-design principles to which they are addressed. The results suggest a tendency towards the development of systems aiming to improve authentication processes and privacy preserving methods.

### 4.2. Architectures of the Future

Deep-learning methods might be suitable to improving safety and efficiency in several data-driven ITS scenarios, as highlighted in [41]. For example, the use of cameras using deep neural networks with positioning at a strategic place (e.g., railway crossing or passenger areas at stations) is an efficient way to evaluate threatening situations. Detected risks can be sent to the corresponding authorities. Moreover, the development of ML techniques to handle the large volume of data generated by sensors [6] can facilitate decision-making processes in today’s railway industry. In this context, the adoption of intelligent monitoring systems, in particular, the vertical-acceleration behaviour of railway wagons, was studied in [59]. Additionally, [60] provided a ML technique based on historical data that is able to overcome uncertainty to improve safety at railway stations.

The speculative contexts related to the volatility of some cryptocurrencies and the high energy consumption in mining processes have generated some ambiguity on the use of blockchain [61]. However, the use of this technology to increase efficiency and security in railways might provide traceability and time and cost efficiency when managing vast amounts of data. Moreover, the distributed nature of this technology will bring resiliency to the system. For example, the effect of a compromised node can be limited to a local area, and therefore, the consequences of potential attacks too [62]. Likewise, the implementation of decentralised architectures can decrease response times of data management systems. In this scenario, the node responsible to provide data can be chosen based in variables such as distance, load and performance. The access to the blockchain can be granted with a set of policies, and transactions can be executed by smart contracts aiming to facilitate, verify and negotiate the contract agreements among different parties.

The appropriate implementation of a cloud computing architecture presents several advantages in the railway industry. Its use can provide efficiency, unification and safety to the train transport services. However, the application of convenient policies is a key challenge to ensure the integrity, confidentiality and availability of the data. From a technical perspective, encryption schemes using novel approaches such as chaotic cryptography can be proposed to protect data integrity [51]. Furthermore, the increasing demand of HSR to connect cities poses significant challenges. For example, the frequent handovers existing in trains running at high speeds might lead to problems in data transmission. Thus, with the aim of improving the safety and efficiency of HSR, novel cloud-computing-based architectures are expected to gain importance, as observed in [19]. The protection level of data (high-level vs. low-level encryption) is chosen depending on the information’s sensitivity. Moreover, cloud architectures might be interesting alternatives not only to increase safety and efficiency at specific scenarios (e.g., HSR), but also to share responsibilities in providing data security [63]. The articles aiming to provide solutions to the topics listed in Section 3.1 are enumerated in Figure 7. Although the distribution does not demonstrate a clear bias towards a specific area, the improvements in communication systems, along with the development of applications using AI, are expected to generate higher interest in the railway industry.

### 4.3. Privacy Challenges

The increasing use of mobile devices; the already mentioned technologies (artificial intelligence, blockchain, cloud, etc.) that aim to improve travellers’ safety and systems’ efficiency; and the massive generation of data brought on by the advent of IoT systems in train stations, raise a number of privacy issues and challenges. Thus, these challenges must be addressed from social, technical and legal and ethical perspectives.

From a technical perspective, the high mobility of vehicles may pose significant challenges to access control, authentication and authorisation procedures [19]. In this context, the frequent handovers existing in high-speed trains should foster the development of fast authentication methods (e.g., [44]). Moreover, the compression of the large volumes of data generated by ticketing systems, sensors, video cameras or other Internet sources can not only bring about more efficient energy consumption systems [64], but also protect passengers’ privacy [65]. Furthermore, with the aim of communicating with other sources to improve safety and provide better services, railways are periodically sending information (e.g., location coordinates) to a network shared with other users. Although these networks are still vulnerable to cyberattacks [66], the implementation of differential privacy [67] or anonymisation techniques [68,69] can be seen as an interesting solution to protect data privacy. Likewise, the development of architectures aimed at dealing with sensitive data should be developed following security and privacy-by-design principles, namely, accountability, authentication, availability, confidentiality, integrity, non-repudiation, revocation and data privacy [41]. Moreover, the implementation of anonymisation techniques would provide compliance to privacy regulations, such as the General Data Protection Regulation (GDPR). Additionally, pseudonymisation techniques to preserve privacy in online communications, or the implementation of microaggregation models for the management of historical data can be seen as interesting solutions to bring privacy to railway services.

Researchers and policy makers have undertaken efforts to accommodate technical railway requirements, global regulations and high security demands. However, the massive volume of data generated by passengers and systems requires one not only to deploy suitable technologies but also to address complex social and ethical challenges, such as user perceptions of consent, accountability and transparency [37]. Thus, security improvements must be followed with the positive perceptions of users with regard to their privacy, liberty and civil rights [48].

### 4.4. Cybersecurity Frameworks and Standards

Railway entities implement multiple approaches when dealing with risk management. Although the differentiation between information technology (IT) and operational technology (OT) systems is not trivial, the correct differentiation between these areas remains a key factor to solving most of the challenges brought on by the digitalisation of the industry. On the one hand, the NIS Directive [55], the NIST cybersecurity framework (CSF) [70] and the ISO27000 family standards (i.e., 27001, 27002 and 27005) [71] can be considered the most relevant standards or frameworks for risk management in IT systems. On the other hand, the more specific methods required in OT systems are provided by ISA/IEC 62443 [72]. These series of standards address the security requirements of industrial automation and control systems (IACS) throughout their lifecycles. Moreover, ISA/IEC 62443 applies to the recently released standard CLS/TS50701, 2021 [73], which is aimed at keeping the security risks of railway systems at acceptable levels.

In accordance with these standards, the already mentioned report published by ENISA [57] defines a control list mapping the NIS Directive with the ISO27001, NIST CSF and CLC/TS50701. Although the stakeholders are not obliged to implement all the measures on the list (which will vary depending on the case), they will be requested to comply with national guidelines and regulations. Related to the railway sector also is that [74] provides a comprehensive review to help the industry adopt the appropriate standard or framework based on the cybersecurity requirements.

With the aim of promoting the effectiveness of the European rail industry, Shift2Rail [75] seeks to bring innovative and market-driven solutions to overcome the several challenges associated with the industry. Among other objectives, the project proposes a generic approach to perform a risk assessment that includes the specific attacker and threat landscapes, the targets’ estimation security level and a detailed procedure for risk assessment (based on IEC 62443). Additionally, the EU-funded project CYRail [76] presented a guide with the most relevant threats targeting railway systems, enabling, therefore, the implementation of more effective cyberdefences.

### 4.5. Training and Awareness

The development of the digitised railway industry is increasing the number of connected devices and leading to more complex infrastructures. These devices open up new attacks vectors for cybercriminals to use to access the railway data. Thus, there is a need to implement effective training and awareness campaigns to prepare the cybersecurity workforce to support railway infrastructure and efficiently defend against cyberattacks [30]. Since these campaigns have been considered from a very generic perspective, further work should consider observing the psychological profile of each individual and provide, therefore, more effective education [77]. Moreover, the education efforts should be followed by the implementation of policies aiming to help personnel apply the proper measures in cases of cyber incidents.

Besides the need to invest resources in promoting better training campaigns for the railway workforce, the several cyber risks existing in train stations should also be communicated to passengers. Thus, accessing public wireless networks or sharing private information (e.g., location) might also be a source of harmful cyber incidents. Relatedly, the development of models to guarantee passengers’ privacy protection will improve the resilience against cyberattacks [40]. Although travellers’ data can be used by railway companies to improve their safety, security and QoS, it can also lead to complex social and ethical challenges in terms of accountability and transparency [37]. Relatedly, the development of global and trusted frameworks to guarantee passengers’ privacy will greatly improve passenger’s perceptions of train services.

## 5. Limitations

Despite the fact that we have followed a sound methodology, our research is subject to some limitations. The literature search was conducted using the Web of Science only. Applying the same search methodology to other databases, such as IEEE Xplore and Scopus, might retrieve other articles (not indexed by the Web of Science) that could be relevant to the topic too. Additionally, our inclusion/exclusion criteria might have limited the collection of potentially interesting articles. In particular, the quality-based criterion, $IC_{4}$, considers only articles published in top journals or which are relevant from a citation perspective. By relaxing this criterion, more articles would be screened. Finally, since the scope of the article was to cover any security and privacy aspect related to the digital universe (i.e., cybersecurity), any research addressing these aspects from a non-technological side (i.e., personnel and structural elements) has been excluded.

## 6. Conclusions and Further Work

Cybersecurity threats, with a particular emphasis on attacks targeting critical infrastructures, are on the rise [78]. While the number of phishing attacks is hitting new records, the transportation industry is already suffering an increase in ransomware attacks [79]. People and vulnerable devices are generally the main attack vectors for perpetrators [80]. To reverse this upward trend, massive investment in railway cybersecurity is taking place, which is expected to be a 16.7 billion dollar market by 2028 [81]. In this context, the evolution towards intelligent railway transportation systems will undoubtedly usher in more effective, efficient and sustainable societies. Likewise, railway infrastructures, including wagons, platforms, stations and rails, will be augmented with the proper sensing and communication technologies so as to collect large volumes of data in real-time and transfer them through fast data networks to data centres for further processing.

Despite the potential benefits, the addition of these technologies introduces novel challenges related to cybersecurity aspects, especially from information security and data privacy perspectives. For instance, sensor devices within the scope of IoT or the emergence of WSNs are suitable technologies to improve decision-making processes due to being able to capture, transmit and analyse humongous amounts of data. However, the limited computational capabilities of these technologies usually hinder the addition of a security layer, and attackers could exploit vulnerabilities to steal data or infiltrate themselves into the sensor network. Likewise, vulnerable devices in railway infrastructures, such as balises, ticketing systems and travel information display systems, can be targets of attack vectors, through which criminals can perpetrate their malicious actions and disrupt their proper functioning. Additionally, the tendency towards AI-based models will not only help predict menacing situations in specific cyber-physical scenarios, but also improve data management to facilitate decision-making processes. Notwithstanding, these models could be maliciously poisoned or biased in favour of malicious actors. Given the ever-evolving and high-complexity nature of railway infrastructures, the management of information security and privacy aspects is far from straightforward, and the development of efficient, scalable, secure, ethical and compliant data architectures is extremely challenging.

To shed some light on this topic, this article has provided a comprehensive review of the knowledge on the railway industry regarding both information security and data privacy perspectives. To this end, we have conducted a timely review of the state of the art in this domain following a strict review methodology. For the sake of structure, studies were classified into three groups according to their scope and goals, called (i) enhanced systems for increasing safety and security in railways; (ii) cybersecurity issues and challenges in railways; and (iii) users’ cybersecurity awareness in railway infrastructures. These studies have been thoroughly described to the readers, by carefully emphasising their security and privacy connotations. Finally, we have also elaborated on a complete discussion related to the open challenges and issues in the field, not only from a technical dimension, but also from societal and ethical perspectives. Hence, the most relevant procedures, architectures and technologies aimed at increasing the cyber resilience in railway transportation systems have been stressed.

This article paves the way to enabling practitioners and stakeholders to leverage horizontal strategies to fill in the identified gaps quickly and accurately. Promising research lines that deserve attention from the scientific community include, but are not limited to:Developing, implementing and deploying IoT devices and WSNs that contain a security layer to minimise the success of malicious attacks. The evaluation of the added overhead, in terms of time and cost, will need to be studied.Integrating novel, emerging communication technologies. To prevent using old, vulnerable communication technologies, the latest trends on the topic, including low-power communication, need to be carefully examined. The impact of this migration requires attention, not only from the security perspective, but also in terms of efficiency and cost.Embracing AI techniques with security-by-design and privacy-by-design for future-ready operations. Since many decision-making systems are AI-based, the incorporation of security and privacy requirements is mandatory to protect not only the railway services, but also the users. Moreover, AI-based techniques could be used to detect anomalous situations and anticipate incoming attacks.Migrating classical data architectures to scalable cloud environments to increase the security of complex applications by sharing responsibilities, e.g., computational capabilities and data security. Consideration of big data technologies with a focus on security and privacy aspects will be desirable.Integrating and deploying blockchain-based applications to increase security and limit the impacts of cyberattacks, while bringing resiliency, transparency and data traceability within the system. In this context, using smart contracts might set the conditions to access the blockchain and establish the rules for performing communication among different parties.Fostering standardisation initiatives to overcome the alarming current lack of interoperability. Defining common ontologies, protocols and standards is required to unify railway systems, efficiently manage data and hence, homogenise cyberdefence procedures. Relatedly, the adoption of the already-mentioned CLC/TS 50701 standard might be a promising move towards the harmonisation of good practices.Developing complete guidelines for risk assessment and risk mitigation in railways. Understanding potential attacks, assessing vulnerabilities and applying the corresponding corrective measures is key to avoiding catastrophic consequences.Balancing open and comprehensible data policies with privacy-preserving mechanisms, including the fulfilment of user consent and the ubiquitous nature of intelligent railway transportation services. Additionally, developing information security standards is suitable.Educating people on the right use of technology by means of personalised training campaigns targeted at users. These initiatives should clearly make users aware of the cybersecurity concerns and provide the necessary skills to use prevention and mitigation techniques when deemed convenient.Make policy makers aware of the need to foster efficient regulations to handle information properly, while always considering privacy and ethical aspects.

## Figures and Tables

**Figure 1 sensors-22-07698-f001:**
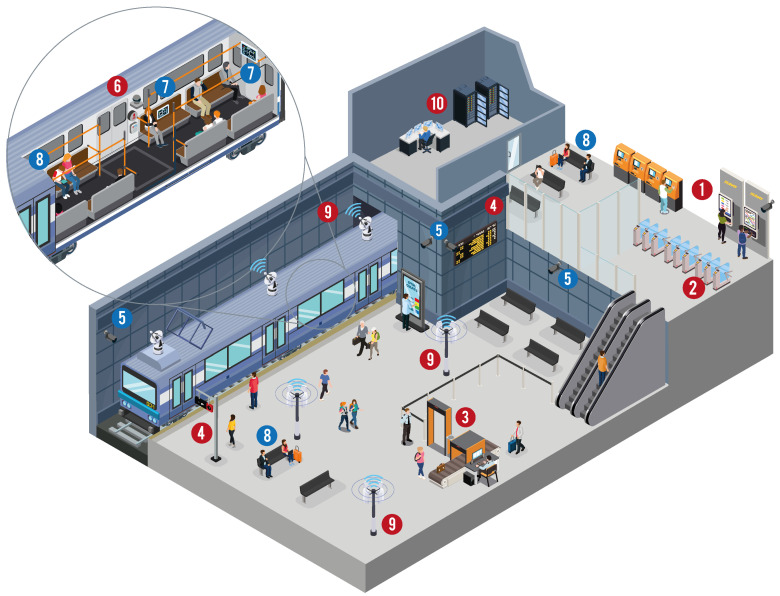
A railway transportation scenario with potential information security and privacy risks. Each number refers to a case described in the full text. Numbers coloured in red are related to information security concerns, and numbers coloured in blue are related to data privacy concerns.

**Figure 2 sensors-22-07698-f002:**
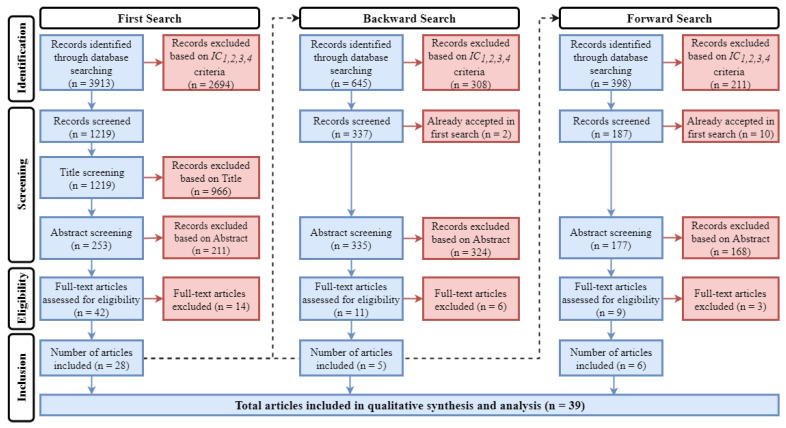
Literature search evaluation methodology.

**Figure 3 sensors-22-07698-f003:**
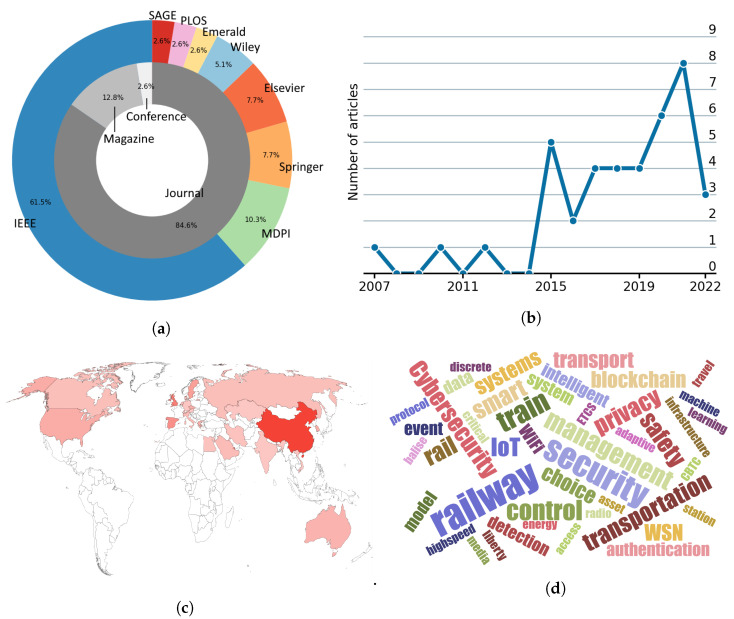
Quantitative analysis of the selected articles in this literature review. (**a**) Distribution of the articles per source type and publisher. (**b**) Temporal distribution of the articles per year. (**c**) Geographical distribution of the authors per their first institution country. (**d**) Word cloud of the most relevant keywords.

**Figure 4 sensors-22-07698-f004:**
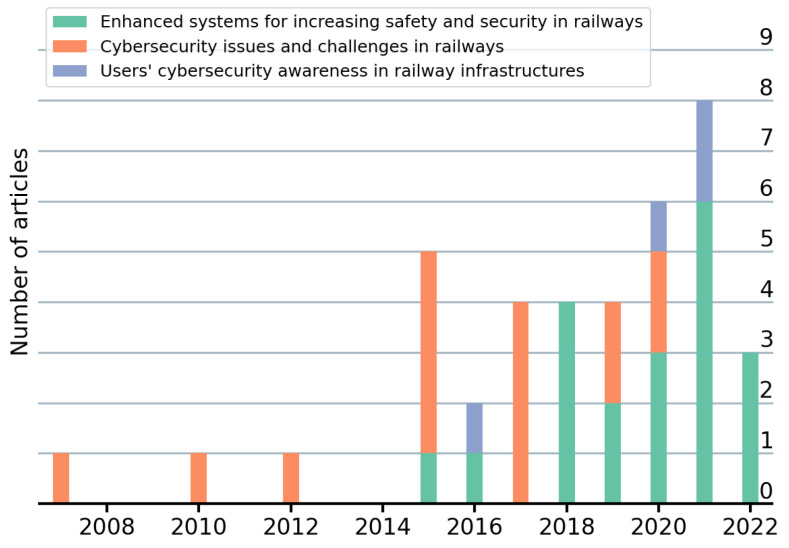
Temporal distribution of the articles per group.

**Figure 5 sensors-22-07698-f005:**
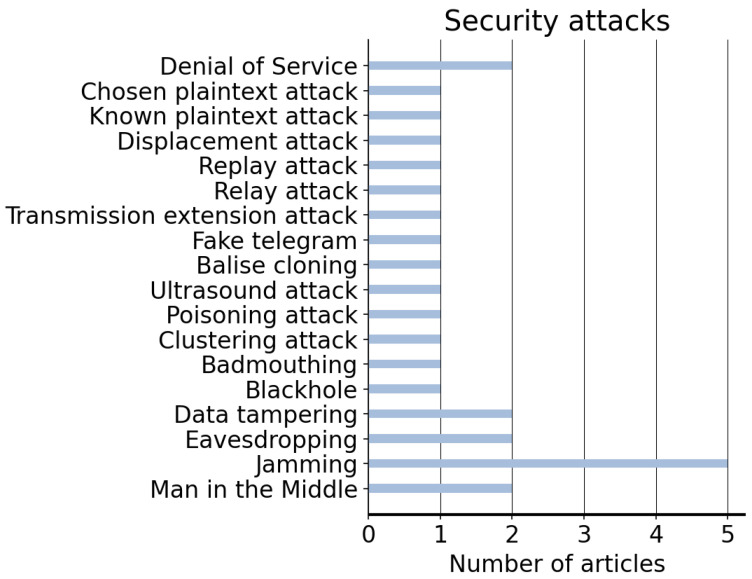
Security attacks to which the described articles are addressed.

**Figure 6 sensors-22-07698-f006:**
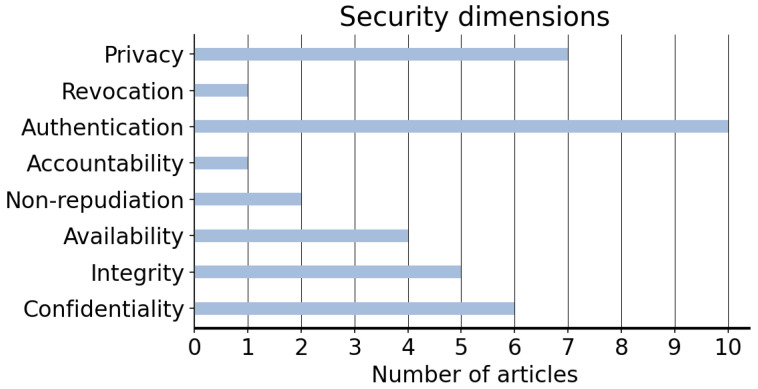
Security dimensions to which the described articles are addressed.

**Figure 7 sensors-22-07698-f007:**
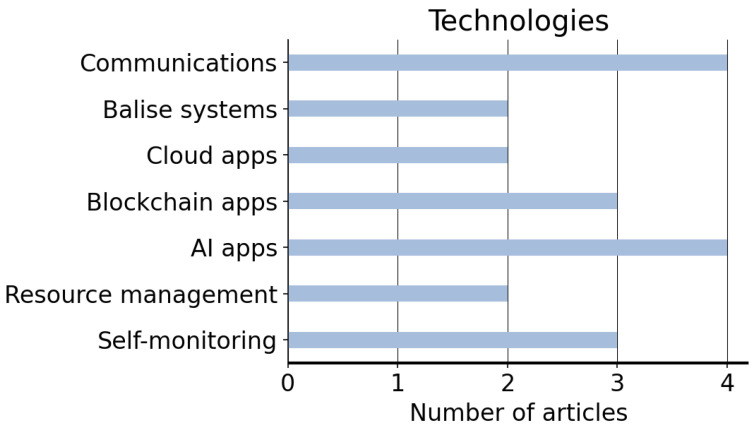
Technologies which the described articles address.

**Table 1 sensors-22-07698-t001:** Research questions addressed in this literature review.

ID	Research Question	Objective	Discussion
RQ1	Which are the current procedures and tools to address the information security and privacy aspects of railway transportation?	The purpose is to summarise the current instruments to digitally protect all the actors involved in railway transportation.	Section 3.1
RQ2	Which are the main challenges that have been identified for making railway transportation more secure and private?	The aim is to collect, organise, classify and summarise the main challenges found in the literature for further discussion.	Section 3.2
RQ3	Are users aware of the security and privacy aspects involved in railway transportation?	The goal is to assess whether society, according to the analysed literature, is aware of the potential security and privacy issues of their railway transportation usage.	Section 3.3
RQ4	Are current practices efficient enough to counter the sophistication of cyberattacks?	The goal is to verify whether current procedures and technological solutions are sufficient to efficiently fight cyberattacks.	Section 4
RQ5	Which technologies or strategies could be used to deal with the identified challenges?	The purpose is to provide a fruitful discussion to improve railway transportation from an information security and privacy perspective.	Section 4
RQ6	Which issues remain open?	According to the knowledge extracted from the literature, the goal is to pinpoint the main limitations in the field to set the ground for further research.	Section 4 and Section 6

**Table 2 sensors-22-07698-t002:** List of the selected articles in this literature review.

Reference	Title	Source	Year	Search
Ai et al. [13]	Future Railway Services-Oriented Mobile Communications Network	IEEE Communications Magazine	2015	Backward
Alawad et al. [14]	Wireless Sensor Networks: Toward Smarter Railway Stations	Infrastructures	2018	First
Bellini et al. [15]	Cyber Resilience Meta-Modelling: The Railway Communication Case Study	Electronics	2021	First
Cabalquinto et al. [16]	It should allow me to opt in or opt out: Investigating smartphone use and the contending attitudes of commuters towards geolocation data collection	Telematics and Informatics	2020	First
Chernov et al. [17]	Security Incident Detection Technique for Multilevel Intelligent Control Systems on Railway Transport in Russia	Proc. Telecommunications Forum	2015	First
Daly et al. [18]	Using ordered attitudinal indicators in a latent variable choice model: a study of the impact of security on rail travel behaviour	Transportation	2012	First
Dong et al. [19]	SVCC-HSR: Providing Secure Vehicular Cloud Computing for Intelligent High-Speed Rail	IEEE Network	2018	First
Duan et al. [20]	Optimal Scheduling and Management of a Smart City Within the Safe Framework	IEEE Access	2020	First
Falahati et al. [21]	Improve Safety and Security of Intelligent Railway Transportation System Based on Balise Using Machine Learning Algorithm and Fuzzy System	International Journal of Intelligent Transportation Systems Research	2022	Forward
Farooq et al. [22]	Radio Communication for Communications-Based Train Control (CBTC): A Tutorial and Survey	IEEE Communications Surveys & Tutorials	2017	Backward
Figueroa-Lorenzo et al. [23]	Alarm Collector in Smart Train Based on Ethereum Blockchain Events-Log	IEEE Internet of Things Journal	2021	First
Fraga-Lamas et al. [24]	Towards the Internet of Smart Trains: A Review on Industrial IoT-Connected Railways	Sensors	2017	First
Hatzivasilis et al. [25]	SPD-Safe: Secure Administration of Railway Intelligent Transportation Systems	Electronics	2021	First
Hodge et al. [26]	Wireless Sensor Networks for Condition Monitoring in the Railway Industry: A Survey	IEEE Transactions on Intelligent Transportation Systems	2015	Backward
Jang et al. [27]	Control of interior surface materials for speech privacy in high-speed train cabins	Indoor Air	2017	First
Kim et al. [28]	Cyber-Physical Vulnerability Analysis of Communication-Based Train Control	IEEE Internet of Things Journal	2019	First
Kour et al. [29]	eMaintenance in railways: Issues and challenges in cybersecurity	Proc. Institution of Mechanical Engineers, Part F: Journal of Rail and Rapid Transit	2019	First
Kour et al. [30]	Cybersecurity workforce in railway: its maturity and awareness	Journal of Quality in Maintenance Engineering	2021	First
Lazarescu et al. [31]	Asynchronous Resilient Wireless Sensor Network for Train Integrity Monitoring	IEEE Internet of Things Journal	2021	First
Lopez et al. [32]	Cyber Security Analysis of the European Train Control System	IEEE Communications Magazine	2015	First
Ma et al. [33]	Interference Control for Railway Wireless Communication Systems: Techniques, Challenges, and Trends	IEEE Vehicular Technology Magazine	2020	First
Mcmahon et al. [34]	Requirements for Big Data Adoption for Railway Asset Management	IEEE Access	2020	First
Moreno et al. [35]	A Survey on Future Railway Radio Communications Services: Challenges and Opportunities	IEEE Communications Magazine	2015	Backward
Mu et al. [36]	Policy-Driven Blockchain and Its Applications for Transport Systems	IEEE Transactions on Services Computing	2020	First
Patil et al. [37]	Public preference for data privacy – A pan-European study on metro/train surveillance	Transportation Research Part A: Policy and Practice	2016	First
Potoglou et al. [38]	Quantifying individuals’ trade-offs between privacy, liberty and security: The case of rail travel in UK	Transportation Research Part A: Policy and Practice	2010	First
Pouw et al. [39]	Monitoring physical distancing for crowd management: Real-time trajectory and group analysis	PLOS ONE	2020	First
Rao et al. [40]	A privacy-preserving framework for location recommendation using decentralized collaborative machine learning	Transactions in GIS	2021	First
Sikora et al. [41]	Artificial Intelligence-Based Surveillance System for Railway Crossing Traffic	IEEE Sensors Journal	2021	First
Sun et al. [42]	Energy-Efficient Communication-Based Train Control Systems With Packet Delay and Loss	IEEE Transactions on Intelligent Transportation Systems	2016	Backward
Thaduri et al. [43]	Cybersecurity for eMaintenance in railway infrastructure: risks and consequences	Intl. Journal of System Assurance Engineering and Management	2019	First
Wang et al. [44]	Improving the Security of LTE-R for High-Speed Railway: From the Access Authentication View	IEEE Transactions on Intelligent Transportation Systems	2022	Forward
Wu et al. [45]	Position Manipulation Attacks to Balise-Based Train Automatic Stop Control	IEEE Transactions on Vehicular Technology	2018	Forward
Wu et al. [46]	Vulnerabilities, Attacks, and Countermeasures in Balise-Based Train Control Systems	IEEE Transactions on Intelligent Transportation Systems	2017	First
Wu et al. [47]	Situation-Aware Authenticated Video Broadcasting Over Train-Trackside WiFi Networks	IEEE Internet of Things Journal	2019	Forward
Zeng et al. [48]	Protecting transportation infrastructure	IEEE Intelligent Systems	2007	First
Zhang et al. [49]	A Novel Stochastic Blockchain-Based Energy Management in Smart Cities Using V2S and V2G	IEEE Transactions on Intelligent Transportation Systems	2022	Forward
Zhao et al. [50]	Resilient Cooperative Control for High-Speed Trains Under Denial-of-Service Attacks	IEEE Transactions on Vehicular Technology	2021	Forward
Zheng et al. [51]	A Lightweight Authenticated Encryption Scheme Based on Chaotic SCML for Railway Cloud Service	IEEE Access	2018	First

**Table 3 sensors-22-07698-t003:** Classification of the selected articles per group.

Group	List of Articles	Discussion
Enhanced systems for increasing safety and security in railways	[13,14,19,20,21,23,25,28,31,34,36,40,41,42,44,45,47,49,50,51]	Section 3.1
Cybersecurity issues and challenges in railways	[17,18,22,24,26,27,29,32,33,35,38,39,43,46,48]	Section 3.2
Users’ cybersecurity awareness in railway infrastructures	[15,16,30,37]	Section 3.3

## Data Availability

Not applicable.

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
