# Peer review of "Information Security and Privacy in Railway Transportation: A Systematic Review"

_sensors, 2022, doi:10.3390/s22207698_

Round 1
Reviewer 1 Report
// Overview
The paper provides a systematic literature analysis concerning information security and privacy on railway transportation systems. A systematic, transparent and sound methodology has been used. In fact, independent researchers (actually, anyone) can freely download the spreadsheets used in their analysis through their website. The paper is well written.
// Suggestions
I think that it is worth to mention that recently published TS 50701 railway security standard addresses the security management for railway applications. The document (directly or indirectly) specifies both process and system/product requirements for railway operators, system integrators and product suppliers. As said, as design practice, privacy should be promoted. Something about it could be said (line 696 ?).
Figure 1 and the explanation from line 62 provides a good overview of railway threat landscape and exposure. This information can be really useful for security risk asessments and management (usually to be accomplished by railway operators). In this matter, as future work, a complete scenario could be described (from the user perspective) that could be used as input for a generic risk assessment. This would be aligned with the research line proposed in line 700.
This study can provide a baseline for further research. Personally, I would also be curious on the degree of training and awareness of all profiles involved in railway activities.
// Comments
Please, find below some comments:
Major:
* Privacy is a complex topic to addressed, in which legal and ethical issues comes into play. This subject is commonly analyzed in other domains (e.g. philosophy, law and regulations). Personally, I believe that this analysis should be out of scope. In addition, the topic is superficially touched through the paper. I would also delete the term from the papers title.
Minor:
* Instead of societal aspects in 4.3, in my opinion the subsection is just covering training and awareness. Societal aspects cover a much wider area, in which, for example, the security risk appetitive or legislation subjects are considered. I would just change the title of the section to "4.3 Training and awareness."
* The introduction section seems to large. I would split it into several sections (e.g. Introduction, Railway transportation systems, Goal/Objective).
* Line 161: Besides of the traditional CIA tried, other security properties are considered, such as authenticity or accountability. Risk assessment and management methodologies usually provide a formal definition of those properties to be analyzed. I would suggest to check out those methodologies and definitions.
* Line 576: Generally, the use of AI techniques, for example machine learning learning methods, are not accepted for safety-related applications.
Reviewer 2 Report
The overall impression of the technical contribution of the current study is marginal. However, the Authors may consider doing necessary amendments to the manuscript for better comprehensibility of the study.
1. The abstract may focus more on the technical aspects of security and privacy constraints in contemporary techniques.
2. The techniques or privacy policies that are considered in the evaluation process must be briefly discussed in the abstract.
3. systematisation of knowledge (SoK) mentioned in the title and abstract is nowhere elaborated throughout the manuscript. what is SoK then.
4. As intelligent transportation systems is the keyword, at least a single paragraph on this would assist in better understandability of the study.
5. The references are missing in the submitted document, authors must make sure, that the references are adequately provided to make the study evident.
6. Authors must clearly explain the procedure followed in searching the existing studies on security aspects of railway transport, and what are the search repositories that are targetted, for example, Scopus/Web of Science/IEEE. The sub-sub section 2.3 must be tremendously improved. Please list the keywords that are used in the search process.
7. A background section discussing the metrics for the evaluation of various security models may be discussed.
8. The review comprises too many technologies like blockchain, Secured cloud applications, AI-based applications. But please make it clear on what aspects of security the study focus on, like on confidentiality, integrity, authenticity, availability, or any other aspect.
9. What aspects of railway transportation security does the current study target, is the security of data communicated among the stakeholders (passengers/staff/signal data), intra-communication among the railway base-stations and rail engines, sensor data or data associated with ticket reservation platform?
10. what are various cybersecurity frameworks that are considered in the current study.
11. what is the impact analysis of the current study, and how do the conclusions drawn in the current study influence future works.
12. Please add a table discussing various techniques that are being used in enforcing the security aspect of railway transport along with metrics-based evaluation.
13. what are the limitations of the current review, or any aspects that are considered in the current review to confine the study.
14. Authors are recommended to present the impact analysis and future perspectives of the study as a separate sub-section. More statistical analysis is desired in the discussion section of the manuscript.
15. A thorough proofreading of the document is suggested as it has grammatical and typo errors.
Round 2
Reviewer 2 Report
The overall impression of the technical contribution of the manuscript is Promising. However, the authors need to do minor amendments for better comprehensibility of the study.
1. Authors have to provide more statistical data for better comprehensibility of the data. Too much inadequately summarised text-based discussion in the revised manuscript. Authors are strongly recommended to add tables or graphs.
2. Impact analysis and future perspectives must be presented as the summary of the current review, rather than at the start of the discussion.
3. In the section "4.4. Cybersecurity frameworks and standards", authors must discuss more technical aspects of the standards, please refer https://doi.org/10.3390/electronics11142181 for a better idea.
4. What cyber attacks are being analyzed from the study, the tools/mechanisms discussed in the study robust against Bluesnarfing, Playback Attack, Enumeration, DoS, and Carryover attacks?
5. Concerning the privacy aspects are the models robust against Anonymity, Ineligibility, and Context-based privacy?
